# The development status and future trends of lubricant additives technology: Based on patents analysis

**Mianqing Wang[1], Hua He[2], Xi Fang[3], Hui Li[4]***

**1** School of Intellectual Property, Shanghai University, Shanghai, China, **2** School of Management, Shanghai University, Shanghai, China, **3** School of China-Europa Intellectual Property, Shanghai Institute of Technology University, Shanghai, China, **4** Office of Scientific Research, Shanghai Technical Institute of Electronics & Information, Shanghai, China

* 865100682@qq.com

**Data Availability Statement:** All relevant data are within the manuscript and its Supporting information files.

**Funding:** The author(s) received no specific funding for this work.

## Abstract

In order to reveal the current status and future trends of lubricant additives, this study analyzes the structured and unstructured data of 77701 lubricant additive patents recorded by Patsnap. The results show that China is the country with the largest number of patents in this field, and the United States is the main exporting country of international technology flow; the current research and development of lubricant additives is dominated by multifunctional composite additives; environmentally friendly additive compositions are the current research hotspot; and more environmentally friendly and economically degradable additives have more development potential in the future. Overall, this study provides a comprehensive understanding of the research and application of lubricant additives and contributes to the future development of the lubricant industry.

## 1. Introduction

In the realm of mechanical engineering, ensuring the efficient and stable operation of moving parts within a mechanical system is paramount. However, friction and wear, inherent consequences of mechanical motion, have historically limited the longevity and effectiveness of mechanical systems [1]. Some studies suggest that around 23% of the world's energy usage can be attributed to the energy consumed by friction and wear in mechanical component interactions [2]. To address this challenge, a common approach is the use of liquid lubricants or other lubrication strategies on mechanical moving parts to minimize energy loss from friction and wear. A study by Holmberg and Erdemir [2] revealed that prolonged use of lubricants (over 15 years) can reduce energy losses from friction and wear by up to 40%. This reduction in energy losses could lead to significant savings, equivalent to about 1.4% of the annual global GDP and 8.7% of the total global energy consumption.. Currently, to keep up with the advancements in mechanical engineering and the evolving requirements of internal combustion engines in cars, the use of different types and functions of additives in lubricating oils (base oils) has become a significant trend. These additives aim to reduce energy consumption by minimizing friction

**Competing interests:** I have read the journal's policy and the authors of this manuscript have the following competing interests: No conflict of interest exits in the submission of this manuscript, and manuscript is approved by all authors for publication.

and wear through enhancements in physical and chemical properties, as well as lubricating abilities. Lubricant additives can be classified as single agents with specific performance characteristics or compound agents that combine multiple single agents to create a multifunctional complex agent with diverse properties. These additives serve various functions such as antioxidants, viscosity index improvers, rust inhibitors, anti-foaming agents, etc. The functional properties of the main types of lubricant additives are detailed in S1 Table. Lubricant additives are used in many important sectors and industries such as transportation, manufacturing, power generation, oil and gas, food, and pharmaceuticals [3]. Thus, they have been described as a "barometer" of industrial development. Typically, the demand for lubricant additives increases with a thriving macro-economy and rapid industrial expansion. This correlation highlights the significance of lubricants and the need for their additives.

In recent years, there has been a growing need for premium lubricants to save energy and safeguard the environment. As a result, the lubricant industry has been focusing more on producing high-end products, leading to a significant increase in the demand for and diversity of additives.. In 2005, the European Union made the decision to provide backing for the implementation of the European eco-label in order to encourage the use of environmentally friendly lubrication products that have minimal impact on the environment [4, 5]. In 2015, the global adoption of the Paris Agreement by 178 countries and regions significantly influenced the research and development of lubricant additives. This led to a transition towards utilizing eco-friendly materials, increasing the longevity of lubricants, and decreasing the frequency of replacements. These changes not only reduce resource consumption but also contribute to environmental protection [6]. In this context, nations are actively seeking sustainable solutions in lubricant additive research and development. They try to achieve technological sustainability by incorporating biodegradable components, streamlining production processes, and minimizing energy use and waste generation. Overall, future lubricant additives will need to comply with increasingly stringent regulatory and environmental standards, which in some cases will involve the mandatory replacement of current lubricant additive compositions with more environmentally friendly and cost-effective lubricants.

Patent literature serves as the largest repository of information worldwide, documenting over 90% of global technological knowledge and representing a measurable output of research and development activities globally.. According to statistics from the World Intellectual Property Organization (WIPO), approximately 90% to 95% of global inventive achievements are accessible through patent documents annually, with roughly 70% of these inventions information solely documented in patent literature [7]. In contrast to academic papers, patent literature combines technical, legal, and economic information, making it a vast and comprehensive information source. Efficiently utilizing the information within patent literature can reduce R&D time by an average of 60% and cut R&D costs by 40% [8, 9]. Through analyzing patent documents in specific technical fields and utilizing visualization tools, we can reveal the strengths, weaknesses, current status, and future trends of a particular technical domain [10]. Patent documents contain both structured and unstructured data. Structured data, such as application country, IPC codes, and filing dates, are easily understood and accessible. On the other hand, unstructured data like titles and abstracts, although rich in implicit information, make it time-consuming to extract specific details information due to their variable formats [11]. There are already some mature analytical tools available to analysis structured data [12], but the analysis of unstructured data is still in its early stages [13]. By leveraging both types of data, patent analysis can provide a comprehensive assessment of technical development status and future trends. This analytical approach has been successfully applied in various fields, such as photovoltaic cells [14], nanoscience and engineering [15], and artificial intelligence [16, 17]. However, the development status and future trends of lubricant

additive technologies based on patent analysis, especially the combination with research hot-spots, have not yet been fully investigated and need more in-depth exploration.

This study reveals the global research statutes and future trends of lubricant additives by employing structured and unstructured data from 77701 patent literature from 1965 to 2023. Specifically, this study encompassed the following aspects: (1) Investigating the spatiotemporal and geographical distribution of lubricant additive patents; (2) Revealing the knowledge flow and distribution within lubricant additive technology, analyzing core technologies and research hotspots in this domain; (3) Analyzing the lifecycle and future development directions of lubricant additive technology from both a technical lifecycle perspective and potential development perspectives. There are some margin contribution as follow: (1) this study aims to analyze the current state and future trends of the lubricant additive industry by utilizing both structured and unstructured data from patent literature. The findings will contribute valuable insights to the fields of patent literature metrology and lubricant research; (2) this study is valuable for gaining a comprehensive understanding of the current state, geographical distribution patterns, and research and development priorities within the lubricant industry. It can assist practitioners in fully comprehending the current landscape of lubricant development and research. (3) the exploration of future trends in lubricating oil additives in this paper also helps to point out the direction for research and development personnel. In conclusion, this comprehensive analysis provides an in-depth understanding of lubricant additive technology for research and application.

## 2. Materials and methods

### 2.1 Data collection

Patsnap Patent Search and Analysis Database contains nearly 161 million patent data from nearly 126 countries/regions around the world, which is a rich and comprehensive patent data-base. Additionally, its powerful patent analysis capability has garnered widespread favour among numerous universities and enterprises, serving as a widely adopted tool for patent anal-ysis and intelligence gathering across various industries. Therefore, we conducted a patent search for lubricant additive technology through the Patsnap database. our data collection pro-cess mainly includes three steps:

Firstly, keywords such as 'lubricant' and 'lubricant additives' were used to retrieve patents, resulting in 81,845 patents. The specific Boolean search formula and search records is shown in S2 Table.

Secondly, filters in the Patsnap database were applied to limit the patent data to the period between 1965 and 2023.

Furthermore, certain patents unrelated to lubricant additives were manually excluded to reduce noise in the dataset.

After this process, we ensure our dataset within a reasonable time frame (from 1965 to 2023), That is, all patent applications related to lubricant additive technology were submitted to the patent office between 1965 and 2023 in our paper. Finally, we obtained a total of 77701 patents and used Patsnap 's analysis capability to perform further analysis.

### 2.2 Patent analysis method

**2.2.1 Data mining.**   Text mining is the process of extracting potentially valuable patterns and knowledge from unstructured text, which can be seen as an extension of data mining and knowledge discovery within databases. However, this task is more complex due to the inherently unstructured and ambiguous nature of textual data [18]. Patent titles and abstracts provide a concise overview of patents, containing crucial information about

technology. In this study, we employed Patsnap analysis tools and manual classification methods to extract and organize keywords and abstracts from a dataset of 77701 patents for further analysis.

**2.2.2 Social network analysis.** Social network analysis, also known as structural analysis, is primarily used to explore the relational structure and attributes within social networks [19, 20]. Its significance lies in its ability to provide precise quantitative analyses of various relationships, offering quantitative tools for developing specific mid-level theories, validating empirical propositions, and bridging the gap between the macro and reality. Patents typically have multiple IPC classification codes, indicating that they belong to several IPC classification groups. By examining the co-occurrence patterns of IPC codes of patented technologies, we can analyze research hotspots and core technologies for a specific subject. This paper utilizes social network analysis to construct IPC co-occurrence networks. In social network analysis, nodal centrality serves as a quantitative measure of nodal attributes and can be classified into various metrics such as degree centrality, closeness centrality, eigenvector centrality, and betweenness centrality [21]. This paper primarily focuses on degree centrality, a key metric chosen for analysis. Degree centrality is a crucial indicator for evaluating node centrality in social network analysis, as it effectively demonstrates the strong correlation between a node and other nodes in the network. The larger the degree centrality of a node, the higher its degree centrality, indicating its greater importance within the network. According to Wasserman and Faust [20], for an undirected graph of n nodes, the degree centrality is calculated as follows:

$$C(X_i) = \sum_{j=1}^{n} k_{ij}(i \neq j)$$

Where, $C(X_i)$ is the degree centrality of node i, $\sum_{j=1}^{n} k_{ij}(i \neq j)$ is the number of connections between the computational node and other nodes besides itself. The NrmDegree is calculated as follows:

$$Cnorm(x_i) = \frac{C(x_i)}{n-1}$$

Where, $Cnorm(x_i)$ is the NrmDegree, the larger its value, the more the node network is connected to other nodes and the higher its importance [22].

**2.2.3 Technology life cycle analysis.** G. S. Alshuler, the founder of TRIZ theory, proposed that technology evolution follow a pattern similar to the growth process of living organisms, supported by core technologies [23]. Analyzing the technology life cycle helps determine its overall. The methods for analyzing and determining the technology life cycle typically include the S-curve method, the patent indicator method, the technology life cycle diagramming method and the TCL calculation method [24]. Among them, the S-curve method is widely used in academic due to its resemblance to biological development, aligning with natural laws [25, 26]. The S-curve model suggests that the growth trajectory of technology shares similarities with human development, progressing through stages of inception, growth, maturity, and decline. In other words, the development of technology starts with a slow growth phase during the inception period, followed by rapid expansion, leading into the growth phase, and ultimately reaching its zenith before entering a decline phase. This growth trajectory is graphically represented by an S-shaped curve, commonly referred to as the 'S-curve'. Leveraging this principle, by analyzing historical development data and applying appropriate linear regression models, it is possible to establish the growth curve of a technology and thus predict when the technology will mature or decline in order to gain an advantage over the competition. S-curves

come in two types: symmetric S-curves (Logistic curves) and asymmetric S-curves (Gompertz curves) [27, 28]. Logistic curves are appropriate when the growth of the subject in question is affected by both the amount of growth that has already occurred and the amount yet to be attained. In this study, the development of lubricant additive technology is influenced not only by the accumulation of existing technology but also by the level of industrial development and environmental protection policies and regulations. Thus, this paper utilizes the Logistic model to analyze patent data of lubricant additive technology to predict its life cycle. The equation of the curve as follow:

$$Y_t = \frac{l}{1 + ae^{-\beta T}}$$

Where $Y_t$ is the cumulative number of patents, a is the slope of the S curve growth, β is the growth rate factor, that is, the time point of the turning point in the S curve; l is the peak value of the S curve, also known as the saturation point, which is defined as [l *10%, l * 90%], and the length of time required for the growth and maturity period t. The meanings of the three parameters are as follows (1) saturation: the maximum utility value generated by the use of a certain technology, i.e., the highest value of the estimated cumulative number of patents. (2) growth time: the time required for 10%~90% of the maximum utility value of a technology, i.e., the time required for the growth period and the maturity period. (3) midpoint: the point of inversion of the S-curve, i.e., the point at which the quadratic differential turns from positive to negative.

**2.2.4 Patent portfolio analysis.** It is possible to determine the current development stage of the technology from the macro level by analyzing the technology life cycle. However, further prediction of the development potential of the technology needs to be combined with the core technology branches of this technology [29]. The prediction of the potential for technological development reflects, to a certain extent, the possibility of specific technological development and the potential scale and application prospects of the technological innovation system. This paper utilizes patent portfolio analysis, referring to prior research [30, 31], to predict the development prospects of lubricant additive technologies. Patent portfolio analysis determines the potential for technological development in a given industry by calculating three-dimensional indicators: technology relative growth rate (RGR), technology relative growth potential (RDGR), and technology research and development (R&D) focus. RGR is calculated as the ratio of the average growth rate of a sub-technology's patent application to the average growth rate of all patent applications in that field. If the ratio is greater than 1, it indicates that the growth of technology patents in this field exceeds the growth of patents for the overall technology, which shows that technological innovation activities are relatively active. On the other hand, if the growth of patented technologies in this field is slower, it indicates that technological innovation activities are relatively inactive. RDGR is a measure of the shift in the growth trend of technological innovation activities in a given field. It is calculated by taking the ratio of the average growth rate of patent applications in the two stages before and after a certain sub-technology, divided by the average growth rate of patent applications in the two stages before and after all technologies in the field. If the indicator exceeds 1, it indicates that the rate of patent growth in the field is accelerating, thus suggesting significant future growth potential for technological innovation activities. Conversely, an indicator lower than 1 implies a relatively small future growth potential for technological innovation activities. R&D is calculated by dividing the number of patent applications in a given technology field by the total number of patent applications across all technology fields. Its value indicates the relative significance of a particular technology field compared to others.

## 3. Results and discussion

### 3.1 Distribution and knowledge flows of patents

**3.1.1 Temporal and spatial distribution of patents.** As a key symbol of technological development and market demand, patent applications also serve as a tangible output of R&D activities, which have important indicative significance [32]. By analyzing the temporal and spatial distribution of patent applications, we are able to reveal the evolution of technology market demand and measure the status and influence of different countries and regions in technology competition. Fig 1 shows the global distribution of lubricant additive technology patent application numbers from 1965 to 2023.Regarding total patent applications, China leads with 12544 patent applications related to lubricant additives, indicating a strong demand in the market. Following closely are the United States (11838), Japan (10797), Germany (5220), Canada (3524), and France (1454), all representing developed industrial economies. This distribution underscores the substantial connection between lubricant additives and industrial economies.

Fig 2 reflects the annual patent application trends among the main countries in lubricant additive technology. Overall, it can be divided into three stages. Before 1990, the number of global patent applications in lubricant additives was relatively scarce but exhibited a gradual growth trend. From 1991 to 2000, patent applications showed a decreasing-then-increasing

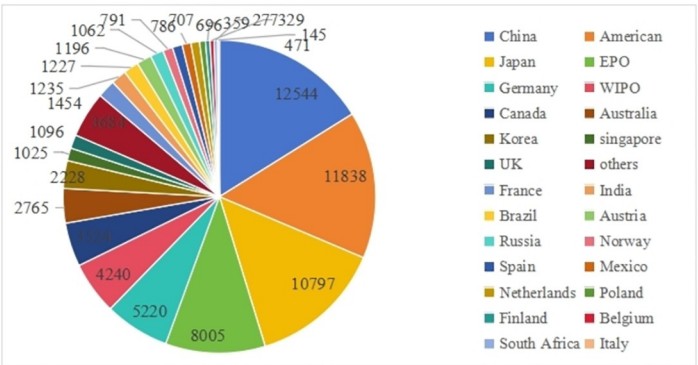

**Fig 1.**

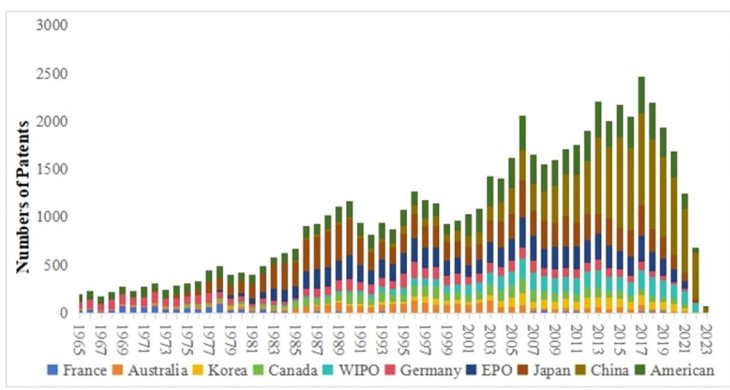

**Fig 2.**

pattern, characterized by fluctuation and subsequent upward movement. However, starting in 2001, there was an explosive growth in global annual patent submissions. Lubricant additives play a crucial role in enhancing various properties of lubricating oils and are influenced by factors such as economic development, industrial structure, and environmental protection policies. Before 1990, lubricant additives were predominantly single-agent products aimed at improving lubricating media. During this period, North America and Europe emerged as key producers and consumers of lubricant additive products, with countries like the United States, Germany, and France leading research and development efforts, gaining expertise in both technological innovation and market expansion. From the 1990s to the late 1990s, as the downstream industrial structure evolved, lubricant additives transitioned from a focus on enhancing functionality to improving economic efficiency, resulting in fluctuating trends in patent applications. Since the early 21st century, the development of lubricant additives has primarily concentrated on energy efficiency, emissions reduction, and improved fuel economy. With the rising demand and increased investment in research and development of lubricant additives by various countries, the number of patent applications has been rapidly increasing.

It is worth noting that the development of lubricant additives technology in China and Japan has made significant progress over the years. Japan initiated research and development in lubricant additives in the 1970s, coinciding with the country's rapid industrial expansion. This surge in industrial activity resulted in a heightened need for advancements in lubricant additives. In contrast, China entered the lubricant additive technology sector at a later stage, with its initial patent applications in this field submitted in 1985. However, starting from 2001, China witnessed a remarkable increase in patent applications related to lubricant additive technology, surpassing all other countries globally. This growth can be attributed to China's expanding market demands post its accession to the World Trade Organization (WTO) in 2001, coupled with its concurrent industrial and economic progress.

**3.1.2 Knowledge flows of patents.**   Fig 3 shows the knowledge flow of patented lubricant additive technologies between countries/regions based on patent citations. The blue squares represent the knowledge inflow countries (i.e., countries of patent backward citations) and the red circles represent the knowledge outflow countries (i.e., countries of patent forward citations). The arrows indicate the direction of the knowledge flow, and the width of the arrows indicates the strength of the knowledge flow (i.e., the number of citations).

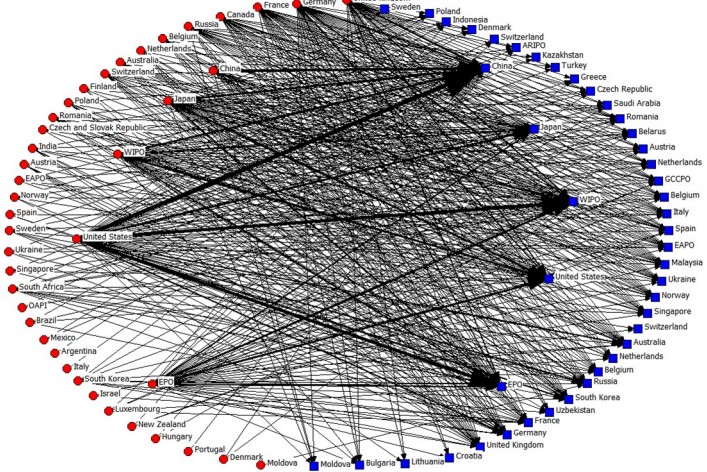

**Fig 3.**

As shown in Fig 3, The World Intellectual Property Organization (WIPO) and the European Patent Office (EPO) are intergovernmental organizations that facilitate the protection of intellectual property through collaboration among States and other international entities. They play a crucial role in connecting global knowledge networks. Technology exchanges are particularly robust among nations like China, the United States, Japan, Germany, France, Canada, the United Kingdom, and Russia. Notably, the United States leads in knowledge dissemination, with its lubricant additive patents being cited 29,108 times by other countries. In comparison, China's patents, despite their large number, received 4,924 citations (only 17% of the US), indicating the superior quality of US patents in this field. China, on the other hand, has the highest knowledge absorption globally, with 21,486 citations of foreign patents related to lubricant additives. This knowledge flow highlights China's reliance on external technical know-how to bridge its technology gap, explaining the rapid growth of its lubricant additives industry despite starting late.

## 3.2 Technological distribution

The International Patent Classification (IPC) code, established by the Strasbourg Agreement in 1971, provides a language-independent hierarchical system of symbols for classifying patents and utility models according to the different fields of technology to which they belong [33]. The IPC codes are an internationally accepted tool for classifying and searching patent documents, and they have made a crucial contribution to organizing, managing, and searching patents [34]. In this study, we counted the IPC codes related to lubricant additive technology and organized the top 10 numbers of records, as displayed in S3 Table. By scrutinizing these IPC codes, we are able to gain comprehensive and systematic insights into the temporal and spatial distribution of lubricant additive technology in different fields, as shown in Fig 4(a) and 4(b).

Fig 4(a) shows the percentage of IPC codes of lubricant additive patent applications in each country. Overall, the proportion of different lubricant additive technologies is more evenly distributed among developed countries, except for China. In contrast, China's lubricant additive patent applications are more concentrated, with only three technologies, C10N30, C10N40 and C10M169, accounting for 70% of the patent applications. In terms of a single technology, all countries have the largest proportion of patent applications in the C10N30, which further indicates the importance of C10N30 in lubricant additive technology. Meanwhile, we can also find that China's patent applications in C10M169 technology are also much more than those of other countries, indicating that China has a strong technological advantage in C10M169 technology. Fig 4(b) shows the development of various lubricant additive technologies in recent decades. It can be seen that in addition to the rapid development of C10N30 and C10N40 technologies, C10M169 technology has also shown a rapid development trend in recent years. The reason may be that China, which started late in the field of lubricant additives, has vigorously carried out the research and development of C10M169 technology on the basis of extensive reference and introduction of foreign advanced technologies.

## 3.3 Research hotspots and core technologies

**3.3.1 IPC code co-occurrence network analysis.** Based on the patent data collected, this study focuses on the co-occurrence frequency statistics of IPC codes related to global lubricant additive technology at the group level. under the framework of social network analysis, we employed the degree centrality calculation formula to determine the degree centrality of IPC codes associated with global lubricant additive technology. Meanwhile, we conducted a visual analysis of the co-occurrence relationship between lubricant additive technology IPC codes by

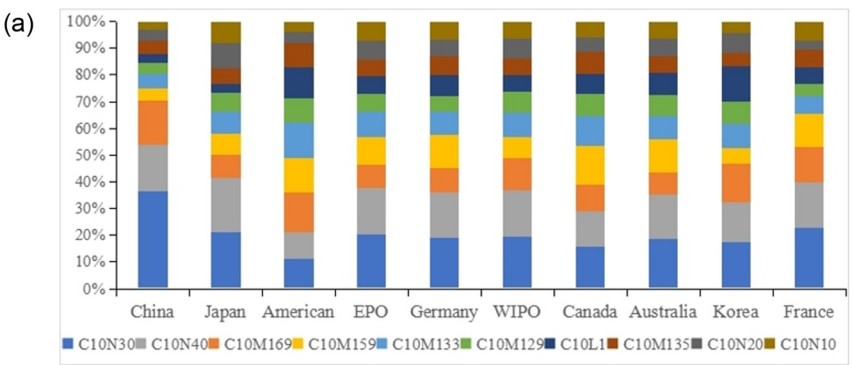

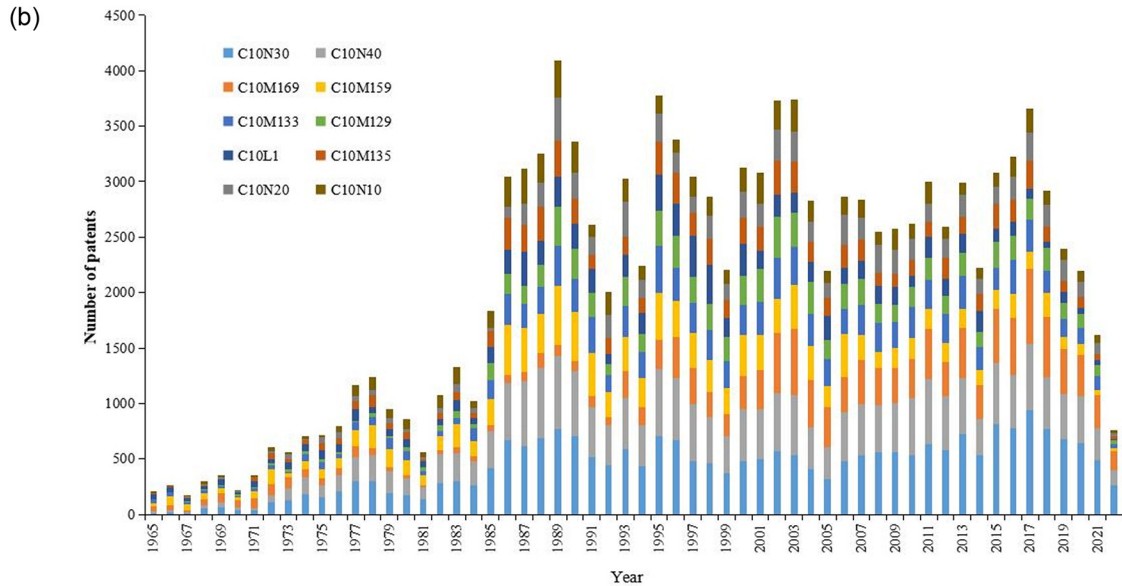

**Fig 4.**

using Ucinet software, as shown in Fig 5. In Fig 5, the size of the circles reflects the degree centrality of the nodes. The larger the circle, the higher the degree centrality value, indicating a greater influence of that node on other nodes in the network. The connecting lines represent the associations between nodes. A connection line indicates that the two nodes co-occur in the same patent document, while the absence of a connection line means that they do not appear together in any patent document. Additionally, we have sorted the nodes in the global lubricant additive technology IPC co-occurrence network according to their value of degree centrality from highest to lowest. Table 1 lists the top ten IPC codes with the highest degree centrality in this network.

Through the analysis of Fig 5 and Table 1, we can clearly observe the close collaboration between different IPC groups in the global lubricant additive technology field. According to the ranking of degree centrality, we can confirm that the IPC classification numbers C10N30, C10N40, C10M159, C10M133, C10M129, C10N10, C10M169, C10N20, C10L1, and C10M135 have the closest connections with other nodes in the entire network and have the greatest influence. This further highlights the concentration of core patents in a few areas within the

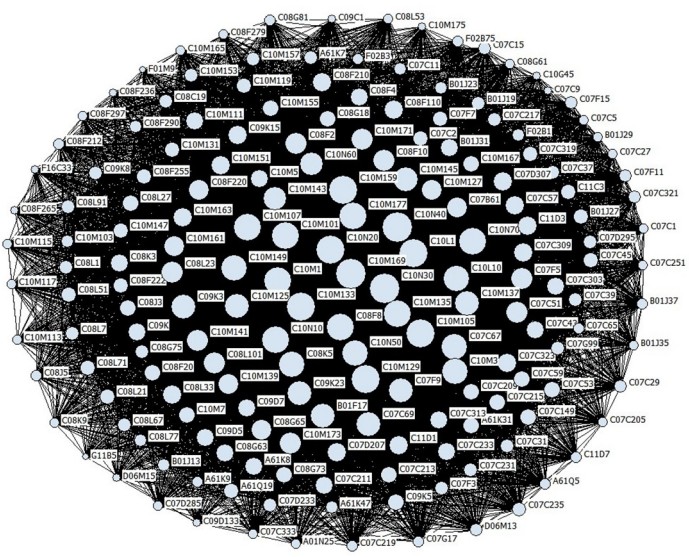

**Fig 5.**

lubricant additive technology field, which is consistent with the development trend of global lubricant additive technology patent applications (as we discussed in 3.2: Technological distribution). It is important to note that the ranking of patent application trends does not align perfectly with the degree centrality ranking of patent IPC co-occurrence networks. This discrepancy is because market demand influences patent application trends, whereas IPC co-occurrence networks offer a more precise depiction of the technological field's core.

**3.3.2 Keywords analysis.** The analysis of keywords in patent literature can assist in identifying the current areas of research in a specific technological field. In this study, we utilized the Patsnap patent analysis software to extract keywords from the titles and abstracts of patents related to lubricant additive technology. Initially, we preprocessed the texts by segmenting Chinese and English words and eliminating common stop words and phrases. Subsequently, we employed the Suffix Tree Clustering algorithm to cluster the texts, considering them as sequences of phrases rather than individual words. This approach enabled us to leverage the contextual information between words and achieve optimal clustering outcomes. Ultimately,

**Table 1. The top 10 values of degree centrality in the co-occurrence network of IPC codes for lubricating oil additive technology patents.**

| Ranking | IPC code | Degree centrality | Nrmdegree |
|---------|----------|-------------------|-----------|
| 1 | C10N30 | 177746 | 0.041 |
| 2 | C10N40 | 148672 | 0.034 |
| 3 | C10M159 | 91671 | 0.021 |
| 4 | C10M133 | 88108 | 0.02 |
| 5 | C10M129 | 76823 | 0.018 |
| 6 | C10N10 | 70422 | 0.016 |
| 7 | C10M169 | 70284 | 0.016 |
| 8 | C10N20 | 67732 | 0.016 |
| 9 | C10L1 | 62485 | 0.014 |
| 10 | C10M135 | 61299 | 0.014 |

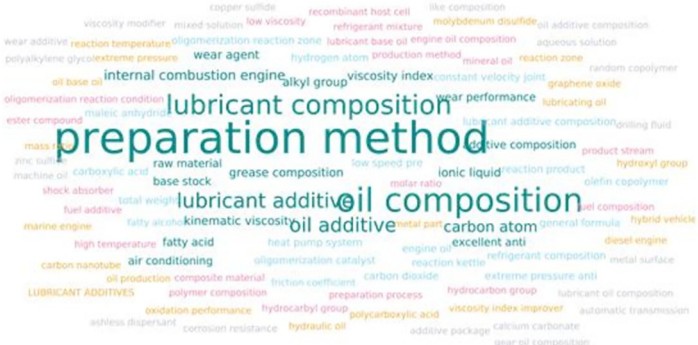

**Fig 6.**

we assigned scores to each cluster and visually presented the top 300 to 500 keywords based on their frequency, as depicted in Fig 5.

Fig 6 demonstrates that the lubricant additive patent technology field commonly utilizes keywords such as "lubricant composition," "composition," "base oil," "compound," "polymer," and others. These keywords indicate that current research focuses on the development of multifunctional compound agents. These agents incorporate single-agent products with distinct functionalities into base oils to meet the lubrication performance requirements of various application scenarios. By controlling the dosage in base oils with specific properties, these compounds not only fulfil the requirements of finished oils but also enable the targeted production of lubricant products with specific quality levels. Moreover, the production process of formulating lubricants using these compounds as raw materials is simple, convenient, time-efficient, and cost-effective. This makes it well-suited to meet industrial needs and efficiently satisfy the demands of the industrial sector.

The keyword cloud also shows frequent terms such as "catalyst", "antioxidant", "detergent", "dispersant", "anti-wear agent", and "viscosity index improver" etc. These terms represent individual components with different functionalities that are extensively used in formulating internal combustion engine oils, gear oils, hydraulic oils, and metalworking fluids. Furthermore, these terms provide valuable information about the components, including "fatty acids", "acrylic esters", "polyolefins", "fatty alcohols", "salicylate salts", and "alpha-olefins" etc. These high-molecular-weight polymers possess the ability to adapt to various environments and meet the performance requirements of lubricants for different products. Importantly, they exhibit excellent thickening properties for base oils.

Additionally, the frequent terms encompass "nanoparticles", "nanosheets", "microorganisms", and "biodegradable" etc., highlighting the increasing focus on environmental protection in the lubricant additive technology field. As a response to environmental initiatives, the field is transitioning towards more eco-friendly and sustainable development. This shift will not only contribute to driving innovation but also foster progress in the industry.

## 3.4 Future trends

**3.4.1 Technology life cycle analysis of lubricant additives.** In this study, we employed Loglet Lab4 to estimate the parameter values of the Logistic model expression based on the collected patent data, with patent application accumulation number used as the vertical axis and application years as the horizontal axis (see Table 2). Furthermore, we draw the life cycle of

**Table 2. Fitting results of the logistic model for lubricating oil additive technology.**

| Parameters | Saturation | Growth time | Midpoint | $R^2$ |
|---|---|---|---|---|
| Value | 128420 | 62.4 | 2015 | 0.999 |

lubricant additives patent technologies based on the trends of patent application accumulation trends of lubricant additives (see Fig 7).

Through the estimated analysis of Table 2 and Fig 7, we can speculate that the saturation value of global lubricant additives patent applications is 128,420, the growth time from the growth period to the maturity period lasted 62 years, and the midpoint occurred in 2015. Based on the midpoint symmetry of the Logistic model, it was possible the inception period of the lubricant additive before 1982, 1983–2015 was the growth period, 2016–2048 was the mature period, and after 2049 it was the decline period. It can be seen that the current global lubricant additive technology is in the mature stage of the technology development cycle. Furthermore, The R2 value of the fitting result of the model is 0.999, which fully proves the reliability and accuracy of our model statistically, and also confirms that the growth of global lubricating oil additive technology patent documents is in line with the logical growth model. Moreover, reviewing the historical development of lubricating additives, we can find that this

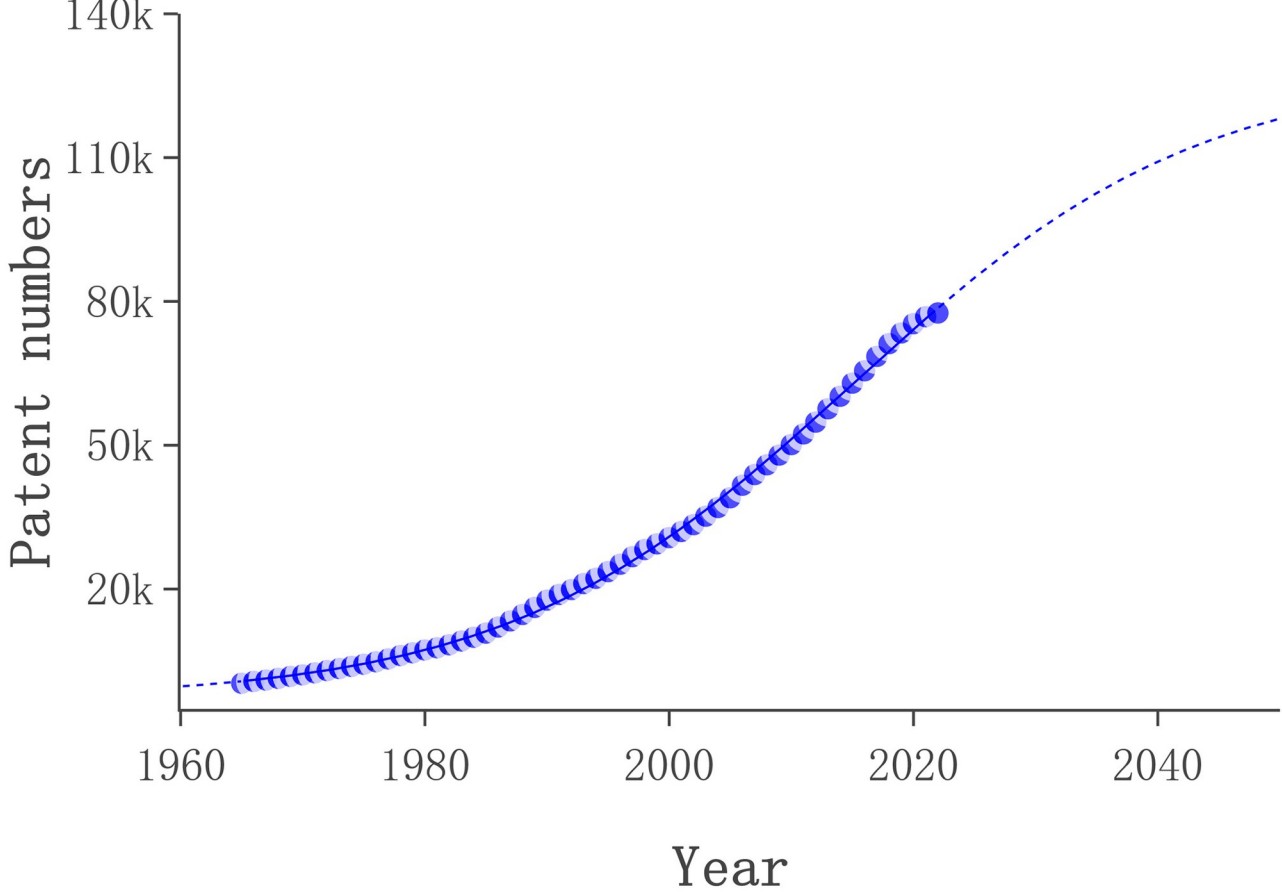

**Fig 7.**

technology was mainly mastered by several developed countries in Europe and North America before 1982, and the technical demand was relatively small, so its research and development progressed slowly. However, from 1983 to 2015, the industrial economies of late-developing countries such as Japan and China have gradually grown, and the demand for lubricant additives has gradually increased, thereby accelerating the research and development process. Especially, the demand of the Chinese market has promoted the progress of lubricant additives after China joined the WTO in 200. In 2015, global attention to environmental protection issues continued to increase, especially since the "Paris Agreement" put forward stricter requirements for the research and development of global lubricant additives, and the market demand for lubricant additives was gradually saturated. Therefore, since 2016, the patent application speed of global lubricant additive technology has begun to slow down, and the technology has entered a mature stage. According to our forecast, by 2049, the technology of lubricant additives will reach the peak of its theoretical development and practical application, and the number of patent applications will decrease year by year, marking that the technology will enter a period of decline. It should be emphasised, however, that when an industry's patented technology enters a time of recession, it does not always imply that the industry itself is in a slump. The S-curve is commonly used to describe the evolution of technology life cycles. When technology reaches a certain level of development, constant optimisation alone will not suffice to fulfil demand but also need to introduce disruptive technology to the industry's evolution. The theory and application of lubricating additive technology have progressively improved, particularly in the context of rising environmental protection standards, and future growth will need the introduction of creative technologies to suit society's needs.

**3.4.2 Development potential forecast of lubricant additive sub-technology.** To further analysis the future growth of the lubricating oil additive sub-technical sector, we anticipate its development potential based on the top 10 IPC codes in terms of patent applications. China was a late starter in the research and development of lubricating additive technology, but the number of patent filings has been quite rapid in recent years, ensuring the accuracy of the forecast. In this study, we performed the patent portfolio analysis to forecast the developing potential of lubricant additive sub-technology by using the patent application submitted after 2000, and data for 2023 was excluded due to the typical 18-month lag in patent publication. When calculating the RDGR, we take 2001–2011 as the before period and 2012–2022 as the after period to calculate, the specific calculation results are shown in Table 3. Meanwhile, we depicted a patent portfolio matrix diagram (shown in Fig 7) based on the results shown in Table 3. There are three dimensions to the patent portfolio matrix graphic. The abscissa represents RDGR, whereas the ordinate represents RGR, and the size of the circle is R&D indicating the relative importance of the technology.

**Table 3. RGR, RDGR and R&D of lubricant additive sub-technologies.**

|  | RGR | RDGR | R&D |
|---|---|---|---|
| C10N30 | 1.05 | -12.30 | 14.7% |
| C10N40 | 0.74 | 10.21 | 11.3% |
| C10M169 | 3.79 | -0.92 | 10.2% |
| C10M159 | -0.50 | 2.56 | 5.2% |
| C10M133 | -0.82 | 4.54 | 6.3% |
| C10M129 | -0.22 | 3.33 | 5.1% |
| C10L1 | 0.44 | -2.57 | 3.7% |
| C10M135 | -0.45 | 3.17 | 4.3% |
| C10N20 | 0.87 | 14.49 | 4.9% |
| C10N10 | 1.01 | 4.43 | 4.0% |

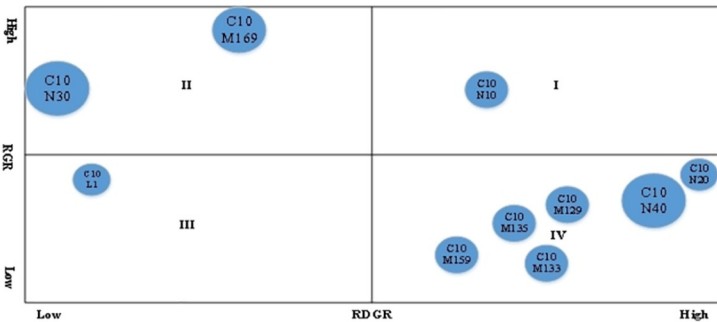

**Fig 8.**

Fig 8 shows that the top 10 sub-technology fields of lubricant additives are scattered in four different areas. Specifically, the C10N10 field is located in Zone I, the C10M169 and C10N30 fields are located in Zone II, the C10L1 field is located in Zone III, and the remaining sub-technical fields are distributed in Zone IV.

The RGR and RDGR are the highest in Zoon I, which represents the development potential is enormous. As a result, the C10N10 field is not only active in technical innovation but also has significant future development potential. However, it should be emphasised that the present investment in C10N10 is quite limited in terms of R&D. This phenomenon implies that there will be plenty of space for expansion in this technology field in the future. Therefore, enterprises and research institutes should pay close attention to this critical technology field.

In Zone II, the technology has reached a relatively mature stage. After a long period of development, the technology has accumulated significantly and secured a substantial market share. Over time, the technology has gradually entered the standardization stage, maintaining a relatively high and stable relative growth rate. Consequently, for the C10N30 and C10M169 fields, although research and development activities remain vibrant, future development faces certain limitations. This phenomenon is probably attributed to oligopoly enterprises having mastered the main technologies in the two technical fields during the long-term technology accumulation process, making it difficult for new enterprises to enter. However, these two fields have relative importance compared to other technical fields from the perspective of R&D. Thus, enterprises and relevant institutions should exercise caution when making investment decisions in these two fields.

Both RGR and RDGR values are quite low in Zone III, indicating that the technology in this zone is in a period of decline, or is about to be eliminated by the market. The main reason for this is that the theory and application of this technology are close to the limit, and the emergence of new technology has led to the rapid replacement of the original technology. Based on RGR and RDGR, the technical growth potential of C10L1 is limited. Combined with R&D, the importance of C10L1 in the field of lubricant additives is also relatively low. Therefore, we can reasonably speculate that C10L1 may not be the key core field of lubricant additives but is more likely to be one of the secondary or mature technology fields, and its future growth potential is relatively limited.

In Zone IV, the RDGR of the technologies is relatively high, while the RGR is comparatively low, indicating that these technologies are in their early developmental stages. Typically, during the initial phases of development, technological growth tends to be slow due to market uncertainty and higher research and development risks, but they possess significant growth potential. Therefore, although the current innovative activities in the fields of C10M159,

C10M135, C10M133, C10M129, C10N40, and C10N20 may not be very active, they hold substantial development prospects for the future. C10M159 is particularly interested in the development of high-performance additives. These additives can sustain lubrication performance by lowering friction and wear under harsh circumstances, hence increasing the service life of mechanical components. The C10M135 is devoted in the application of organic compounds to significantly improve the stability and durability of lubricating oils, which is important in ensuring the lubrication requirements of engines during prolonged high-speed operation, so it has wide applications in the automotive, maritime, and other related industries. The C10M133 is largely concerned with the development of solid lubricants, which significantly minimisig metal-to-metal contact and subsequent wear by forming a lubricating coating during friction. C10M129, C10N40, and C10N20 are more likely to provide innovative additives to meet expanding demands in terms of friction characteristics and wear resistance, providing variety and application opportunities for lubricant additive technology.

## 4. Conclusions

Lubricant additives play a crucial role in various industries and are indispensable for industrial economies. However, the growing global focus on energy conservation and environmental protection has brought about challenges for traditional lubricant additives.. This study examines 77701 patents related to lubricant additives spanning from 1965 to 2023, shedding light on the present status of advancement and forthcoming trends.

First, patent applications in the field of lubricant additives are experiencing consistent growth, indicating a strong market demand. However, there has been a noticeable shift in the spatial distribution of patent applications with a move away from traditional origins in North America and Europe, including countries like the United States, Germany, and France, towards the Asia-Pacific region, particularly China, Japan, and India. China currently leads in the number of patent applications in this field. The United States remains a key player in the technology exchange, boasting a solid technological foundation in lubricant additive technology. Differences in the spatial and temporal distribution of IPC, as well as varying layout emphases in this field, can be attributed to market demand and the industrial structure of each country. Further analyzing the research hotspots and future trends of lubricant additives, it is evident that multifunctional composite additives are receiving widespread attention. The sulfur, phosphorus, and nitrogen compounds traditionally used in lubricant additives are gradually being replaced by non-toxic and tasteless boron compounds, nano compounds, and other alternatives. The future direction of lubricant additive development will prioritize sustainability by employing nanomaterials, graphene, and other degradable components to improve lubrication performance, reduce friction and wear under extreme conditions, and increase cost-effectiveness and eco-friendliness.

The analysis presented in this paper offers a thorough comprehension of lubricant additives' research and application, with valuable implications for the future development of the industry. Nonetheless, there are still some limitations to address. First, the Patsnap database utilized in this paper is a commercial database and is not readily accessible to the public. Second, this study concentrates solely on the analysis of the future status and developmental trends of lubricant additives, with no consideration given to the market competition within the industry. Finally, this paper fails to provide an in-depth discussion of the specific mechanism of lubricant additive technology. Thus, future research can be focused on various aspects of lubricating oil additives. Firstly, exploring the use of specific technologies such as sustainable lubricating oil additives, multifunctional compound additives, alternative additive compounds, and the integration of nanomaterials and graphene in additives from a microscopic

perspective. Secondly, research on lubricating oil additives should consider the relationship with industrial economy, market competition, and industrial structure for a more comprehensive analysis. Lastly, further studies should delve into the specific mechanisms of lubricant additive technology, including the interactions between additives and base oils to understand their impact on reducing friction and wear. Addressing these research directions can advance the field of lubricant additives, enhancing performance, sustainability, and environmental compatibility of lubricant formulations.

## Supporting information

**S1 Table. Types of lubricant additives and their functional properties.**
(DOCX)

**S2 Table. Retrieval formula and results of patents.**
(DOCX)

**S3 Table. IPC codes related to lubricant additive and number of top 10 records.**
(DOCX)

## Author Contributions

**Conceptualization:** Mianqing Wang, Hua He, Xi Fang.

**Data curation:** Hua He.

**Formal analysis:** Hua He.

**Investigation:** Xi Fang.

**Methodology:** Hua He.

**Supervision:** Mianqing Wang, Xi Fang.

**Visualization:** Hua He, Hui Li.

**Writing – original draft:** Hua He, Hui Li.

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
