## [Decision Letter · Decision Letter 0]

21 Feb 2024

PONE-D-23-35055The development status and future trends of lubricant additives technology: based on patents analysisPLOS ONE

Dear Dr. Li,

Thank you for submitting your manuscript to PLOS ONE. After careful consideration, we feel that it has merit but does not fully meet PLOS ONE’s publication criteria as it currently stands. Therefore, we invite you to submit a revised version of the manuscript that addresses the points raised during the review process. Please find the enclosed Reviewer comments about your Manuscript and they recommend for Major Revision on the Manuscript. Therefore, you may Revise the Manuscript based on their comments for furhther consideration.

We look forward to receiving your revised manuscript.

Kind regards,

Venkatasubramanian Sivakumar

Academic Editor

PLOS ONE

Reviewers' comments:

Reviewer's Responses to Questions

**Comments to the Author**

1. Is the manuscript technically sound, and do the data support the conclusions?

Reviewer #1: Partly

2. Has the statistical analysis been performed appropriately and rigorously? 

Reviewer #1: No

3. Have the authors made all data underlying the findings in their manuscript fully available?

Reviewer #1: Yes

4. Is the manuscript presented in an intelligible fashion and written in standard English?

Reviewer #1: No

5. Review Comments to the Author

Reviewer #1: The authors described “The development status and future trends of lubricant additives technology: based on patents analysis”. There are several major as well as minor mistakes in the manuscript. The authors are advised to check all the gaps and errors before its plausible publication in ‘PLOS ONE’.

1. How were the 77701 patents chosen? Were any specific filters applied to the Patsnap database, and if so, could these filters bias the results?

2. What specific methods were used to analyze both structured and unstructured data? Were there any limitations to these methods, and could they have affected the outcomes?

3. Was the Patsnap database chosen for specific reasons? Are there other databases that could have provided complementary information?

4. How were "countries with the largest number of patents" and "main exporting countries" quantified? Are there other ways to measure patent activity that could offer further insights?

5. How were "multifunctional composite additives" defined and identified? Are there other significant research areas that might have been overlooked?

6. How were "environmentally friendly additive compositions" identified as a research hotspot? Are there other metrics that could be used to assess research trends?

7. What specific criteria were used to determine the potential of "environmentally friendly and economically degradable additives"? Are there other factors that could influence their future success?

8. Does this study offer new insights into the lubricant additive field beyond what is already known? Does it contribute significantly to the understanding of research and application trends?

9. Are the findings of this study applicable to different types of lubricants and industries? Are there limitations to the generalizability of the conclusions?

10. The authors are advised to mention the potential future research directions suggested by this study. Are there any specific areas that need further investigation?

11. How does the research and development of lubricant additives compare to other related fields, such as tribology and surface engineering?

12. What are the biggest challenges facing the development of new and improved lubricant additives?

13. What are the ethical considerations associated with the use of lubricant additives?

14. There are so many typos and grammatical mistakes in the entire manuscript, that the authors are advised to rectify all these errors.

6. PLOS authors have the option to publish the peer review history of their article (what does this mean?). If published, this will include your full peer review and any attached files.

Reviewer #1: No

---

## [Author Response · Author response to Decision Letter 0]

7 Mar 2024

Thank you for your valuable comments, we accept all your suggestions, and we have already responded point-to-point in response to the reviewer, please check it.

---

## [Decision Letter · Decision Letter 1]

21 May 2024

The development status and future trends of lubricant additives technology: based on patents analysis

PONE-D-23-35055R1

Dear Dr. Li,

We’re pleased to inform you that your manuscript has been judged scientifically suitable for publication and will be formally accepted for publication once it meets all outstanding technical requirements.

Kind regards,

Venkatasubramanian Sivakumar

Academic Editor

PLOS ONE

Additional Editor Comments (optional):

Since, the Author has revised the Manuscript as per the comments of the Reviewer #1, the Manuscript shall be accepted for publication Plos One..

Reviewers' comments:

Reviewer's Responses to Questions

**Comments to the Author**

1. If the authors have adequately addressed your comments raised in a previous round of review and you feel that this manuscript is now acceptable for publication, you may indicate that here to bypass the “Comments to the Author” section, enter your conflict of interest statement in the “Confidential to Editor” section, and submit your "Accept" recommendation.

Reviewer #1: All comments have been addressed

2. Is the manuscript technically sound, and do the data support the conclusions?

Reviewer #1: Yes

3. Has the statistical analysis been performed appropriately and rigorously? 

Reviewer #1: Yes

4. Have the authors made all data underlying the findings in their manuscript fully available?

Reviewer #1: Yes

5. Is the manuscript presented in an intelligible fashion and written in standard English?

Reviewer #1: Yes

6. Review Comments to the Author

Reviewer #1: The authors have responded all the queries very nicely and subsequently upgraded the manuscript as per the suggestions. So, the manuscript may be accepted for plausible publication in its current form.

7. PLOS authors have the option to publish the peer review history of their article (what does this mean?). If published, this will include your full peer review and any attached files.

Reviewer #1: No

---

## [Editor Report · Acceptance letter]

23 May 2024

PONE-D-23-35055R1 

PLOS ONE

Dear Dr. Li, 

I'm pleased to inform you that your manuscript has been deemed suitable for publication in PLOS ONE. Congratulations! Your manuscript is now being handed over to our production team.

Kind regards, 

on behalf of

Dr. Venkatasubramanian Sivakumar 

Academic Editor

PLOS ONE